# Reducing Application of Nitrogen Fertilizer Increases Soil Bacterial Diversity and Drives Co-Occurrence Networks

**DOI:** 10.3390/microorganisms12071434

**Published:** 2024-07-15

**Authors:** Feng Wang, Hao Liu, Hongyan Yao, Bo Zhang, Yue Li, Shuquan Jin, Hui Cao

**Affiliations:** 1Ningbo Key Laboratory of Testing and Control for Characteristic Agro-Product Quality and Safety, Ningbo Academy of Agricultural Sciences, Ningbo 315040, China; fwang82@163.com (F.W.); yaohongyan2000@163.com (H.Y.); jinshuq@126.com (S.J.); 2Institute of Farmland Water Conservancy and Soil-Fertilizer, Xinjiang Academy of Agricultural and Reclamation Sciences, Shihezi 832000, China; 3Key Laboratory of Agricultural Environmental Microbiology, Ministry of Agriculture and Rural Affairs, College of Life Sciences, Nanjing Agricultural University, Nanjing 210095, China; 2022216010@stu.njau.edu.cn (H.L.); 2021216022@stu.njau.edu.cn (B.Z.); 2022216011@stu.njau.edu.cn (Y.L.)

**Keywords:** nitrogen fertilizer reduction, soil bacterial diversity, nutrient competition, co-occurrence networks, competitive release, soil preservation

## Abstract

Reducing nitrogen fertilizer application highlights its role in optimizing soil bacterial communities to achieve sustainable agriculture. However, the specific mechanisms of bacterial community change under these conditions are not yet clear. In this study, we employed long-term field experiments and high-throughput sequencing to analyze how varying levels of nitrogen application influence the soil bacterial community structure and co-occurrence networks. The results show that reducing the nitrogen inputs significantly enhances the diversity and evenness of the soil bacterial communities, possibly due to the diminished dominance of nitrogen-sensitive taxa, which in turn liberates the ecological niches for less competitive species. Furthermore, changes in the complexity and stability of the bacterial co-occurrence networks suggest increased community resilience and a shift toward more mutualistic interactions. These findings underline the potential of reduced nitrogen application to alleviate competitive pressures among bacterial species, thereby promoting a more diverse and stable microbial ecosystem, highlighting the role of competitive release in fostering microbial diversity. This research contributes to our understanding of how nitrogen management can influence soil health and offers insights into sustainable agricultural practices.

## 1. Introduction

Soil microbial communities are crucial for ecosystem functions such as nutrient cycling, organic matter decomposition, and promoting plant health [1]. Among these microorganisms, bacteria play a significant role due to their diversity and metabolic capabilities. Nitrogen (N) is a key nutrient influencing the structure and function of these bacterial communities [2]. In agricultural systems, nitrogen fertilizers are widely used to enhance crop yields. However, their overuse can lead to detrimental environmental effects, including nutrient leaching, greenhouse gas emissions, and decreased microbial diversity [3,4].

Recent studies have focused on the effects of nitrogen fertilizer inputs on soil microbial communities. These studies have generally found that nitrogen input leads to significant changes in the bacterial community composition and diversity [5,6]. High levels of nitrogen can lead to nutrient imbalances in the soil, negatively impacting microbial diversity [5]. Excessive fertilization can result in certain bacterial groups dominating in nitrogen-rich environments, often at the expense of more diverse microbial communities [7]. This may reduce the soil’s ability to perform essential functions, as a less diverse community may be less resilient to changes and disturbances [8]. However, despite these findings, the specific mechanisms driving these changes are not well understood, which limits our ability to effectively manage the impacts of nitrogen reduction [9]. The mechanisms behind these changes are likely highly correlated with alterations in the soil nutritional status. In conditions where soil nutrients are in excess, certain bacterial taxa proliferate rapidly due to nutrient preference, whereas their abundance may decline with a reduction in nutrients [10]. The phenomenon of competitive release plays a pivotal role in this process; following the reduction or removal of dominant competitors, subordinate species may flourish [11]. Within soil bacterial communities, when the nitrogen input is decreased, the abundance of dominant nitrogen-sensitive taxa declines, thereby liberating niches and resources that allow other bacteria with lower nitrogen dependency to proliferate [12,13]. This results in shifts in both the composition and diversity of bacterial communities. Competitive release has been theoretically recognized, but few studies have explicitly investigated its occurrence and implications in soil bacterial communities under reduced nitrogen conditions [9].

The impact of nitrogen reduction on soil microbial co-occurrence networks and its profound implications for microbial community dynamics is a critical research area, particularly when exploring interactions among microbial taxa and ecosystem functions [14,15]. The complexity and stability of these networks reflect the ability of microbes to adapt to environmental changes, especially in terms of nutrient fluctuations [16,17]. Current research has shed light on some aspects of the impact of nitrogen fertilizers on microbial communities [18,19]; however, the changes in the relationship between competitive release mechanisms and bacterial co-occurrence networks following nitrogen reduction remain to be further studied. A deeper understanding of this mechanism is crucial for revealing how nitrogen reduction can alter ecosystem functions by influencing microbial interactions and network dynamics. This not only enhances our knowledge of microbial community resilience under nitrogen-limited conditions but also provides key insights for developing more sustainable nitrogen management strategies in agricultural ecosystems.

Therefore, this study aims to fill these gaps by exploring the specific mechanisms by which nitrogen fertilizer reduction affects the soil bacterial community structure and co-occurrence networks. Through long-term experimental plot simulations and high-throughput sequencing technology, we aim to elucidate the dynamics of soil bacterial communities under different nitrogen fertilization regimes. By elucidating the mechanisms of competitive release and its implications for the bacterial community structure and co-occurrence networks, our findings will contribute to a better understanding of soil microbial ecology under a changing nutrient level.

## 2. Materials and Methods

### 2.1. Site Description, Experiment Design and Sample Collection

A four-year field experiment (from 2018 to 2022) was conducted at the research base of Ningbo Academy of Agricultural Science, located in Hengxi Town, Ningbo City, Zhejiang Province, China (29°40′ N, 121°35′ E). This region experiences a humid subtropical monsoon climate with maritime characteristics due to the influence of the East China Sea. The annual average temperature is 16.2 °C, and the average annual precipitation is 1555 mm. The soil type is acidic red soil derived from Quaternary red clay, classified as Udic Ferralsol in the Chinese soil taxonomy and as Acrisol in the FAO soil classification system [20].

In this study, we established four-year fixed-location experiments with the following nitrogen application rates: (1) N800: N fertilization rate of 800 kg/ha, local conventional fertilization; (2) N600: N fertilization rate of 600 kg/ha, reduce N amount by 25%; (3) N400: N fertilization rate of 400 kg/ha, reduce N amount by 50%; (4) N0: no N fertilization, reduce N amount by 100%. Cucumbers, with two cropping cycles per year, were planted in all the experimental plots. The nitrogen fertilizer application rates were adjusted using urea (N content 46.4%), which was applied before planting, with two applications each year. Apart from the nitrogen application rates, all the other agronomic practices were consistent across the plots, as detailed in Appendix A. Soil samples were taken from the top 20 cm of each plot with a stainless steel cylinder (3 cm inner diameter) one week after the harvest of cucumbers in December 2022. Ten soil cores was collected in an S-shaped transect from each plot and then mixed to form one composite sample with a weight of approximately 300 g. The samples were immediately shipped to the laboratory on ice packs, and then sieved through a 2 mm mesh to remove roots and other debris. A subsample of the sieved soils was stored at 4 °C for genomic DNA extraction. The other parts of the sieved soil samples were air-dried to determine the soil physicochemical properties. The rest of the soil samples were stored at −80 °C for future use.

### 2.2. Soil Physicochemical Properties

For each soil sample collected in as Section 2.1, the soil pH, organic matter (SOM), total nitrogen (TN), available phosphorus (AP), available potassium (AK), available nitrogen (AN), ammonium (NH_4_^+^-N) and nitrate nitrogen (NO_3_^−^-N) were measured. The methods for determining these soil physicochemical properties were referenced from previously studies [21,22].

### 2.3. Soil DNA Extraction and 16S rRNA Gene Sequencing

The total soil genomic DNA was extracted from 0.5 g of soil sample using the FastDNA™ Spin Kit (MP Biomedicals, Irvine, CA, USA) according to the manufacturer’s instructions. The extracted DNA was then purified using the PowerClean DNA Purification Kit (PowerClean, New Britain, CT, USA), and its concentration and quality were measured using the NanoDrop ND-1000 spectrophotometer (Thermo Scientific, Wilmington, DE, USA). The V4 region of the bacterial 16S rRNA gene was amplified using the primers 515F (5′-GTGCCAGCMGCCGCGGTAA-3′) and 806R (5′-GGACTACHVGGGTWTCTAAT-3′) [23]. The PCR parameters were as follows: an initial denaturation at 94 °C for 3 min; followed by 24 cycles of denaturation at 95 °C for 5 s, annealing at 57 °C for 90 s, and extension at 72 °C for 10 s; with a final extension at 72 °C for 5 min. The PCR products were purified using the GeneJET™ Gel Extraction Kit (Thermo Scientific, Waltham, MA, USA) and the purified products were sequenced by TinyGene, Ltd., (Shanghai, China) using the Illumina Miseq 2 × 250 bp platform (Illumina, San Diego, CA, USA).

### 2.4. Processing of Sequence Analysis

The raw sequencing sequences were processed using the QIIME pipeline (version 1.9.0) [24]. Initially, quality filtering was performed using Cutadapt (V1.9.1) to remove low-quality sequencing reads shorter than 150 bp or with an average base quality score below 20 to obtain high-quality sequences [25]. The UCHIME method was employed to detect and remove chimeric sequences [26]. Subsequently, the sequences were clustered using the UCLUST function, with a similarity threshold set at ≥97% to generate operational taxonomic units (OTUs) [27]. Finally, the OTUs were annotated using the Mothur algorithm [28]. During the annotation process, information about the representative taxonomic units in each sample was retrieved using the Silva database (https://www.arb-silva.de/ 15 March 2024), and the number of OTUs in each sample was calculated.

### 2.5. Statistical and Bioinformatic Analyses

The R “vegan” package was used to calculate the α-diversity indices for the bacterial communities, including Shannon, Chao1, Simpson and Pielou [29]. The “vegan” package was also used for the principal coordinate analysis (PCoA) based on the dissimilarity matrices, and PERMANOVA analysis was carried out to reveal the β-diversity compositional differences in the bacterial communities by using the “Adonis” command of the “vegan” package in R [30]. The bacterial community homogenization effect between the samples was analyzed using the “betadisper” function in “vegan”. Differences in the bacterial community composition were assessed using the Kruskal–Wallis test, and pairwise comparisons between groups were conducted using the Wilcoxon rank-sum test [31]. Similarity percentage analysis (SIMPER) was performed using the “vegan” package to calculate the contribution of specific taxa to the overall community dissimilarity [32].

We used Spearman’s correlation between the OTU and the soil properties to determine the ecological preferences of major phylum, following the approach of Oliverio et al. [33]. In addition, Mantel tests were performed to investigate the correlation between the soil properties and main bacterial phylum [34]. The structural equation model (SEM) for changes in the soil properties and bacterial community diversity was based on the research of Hou et al. [35].

The microbial networks were established following previous studies [36,37]. We combined the reduced N fertilizer treatments in pairs, and subsequently, divided them into three groups for network construction. Network 1 included the N800 and N600 treatments, Network 2 included the N600 and N400 treatments, and Network 3 included the N400 and N0 treatments. Briefly, Spearman’s rank correlation coefficients were calculated using the “psych” R package, then the feature datasets (*p* < 0.01 and |r| > 0.6) were captured for network analysis, and the network graphs were visualized and topologically characterized (average degree, average weighted degree, average clustering coefficient, modularity, number of modules) using Gephi software (version 0.9.22). The network stability was reflected through the network average degree and natural connectivity, which were calculated by randomly removing a certain percentage of nodes from each network [16]. The keystone taxa in the network, including the peripherals, connectors, module hubs and network hubs, were calculated with reference to previous studies [38].

## 3. Results

### 3.1. N Fertilizer Reduction Significantly Changed Soil Bacterial Community Structure

As shown in Figure 1A, the implementation of a strategy for reducing nitrogen fertilizer application has led to significant changes in the α-diversity of the soil bacterial communities. Specifically, the Chao1, Shannon, Simpson, and Pielou indices have all exhibited a significant increasing trend. This indicates that not only has the diversity of the bacterial communities increased after the reduction of N fertilizer, but the evenness of the communities has also been enhanced. Further analysis based on principal coordinate analysis (PCoA) of the β-diversity has shown (Adonis R = 0.638, *p* = 0.002) that the reduction of nitrogen fertilizer application has resulted in significant changes in the composition of the soil bacterial communities (Figure 1B). With the reduction of nitrogen application, the dissimilarity in the community composition grows, along with an increasing trend toward community heterogeneity (Figure 1C).

Among the ten major bacterial phyla, the abundance has undergone varying degrees of variation, with the abundance of two phyla showing a decreasing trend and seven phyla showing an increasing trend (Figure 2A,B). Notably, the Proteobacteria phylum has experienced a substantial change in abundance, decreasing from 59.68% to 35.56%, and the differences between the bacterial communities caused by the abundance change of Proteobacteria are continuously increasing (Figure 2C). Additionally, similar patterns are found at the genus level, with the abundance of a few genera decreasing dramatically, while the majority of the genera saw an increase in abundance (Appendix A). These results collectively suggest that the reduction of nitrogen fertilizer application may have reshaped the structure of the soil bacterial communities.

### 3.2. Correlations between Environmental Factors and Bacterial Community

Our study findings indicate that a reduction in nitrogen fertilizer application led to a decrease in various forms of nitrogen content in the soil, an increase in the C/N ratio, and a rise in the soil pH (Appendix A). Significant changes in the soil bacterial community structure were observed, which can largely be attributed to alterations in the soil environment following the reduction of nitrogen fertilizer. As shown in Figure 3A, the shifts in the bacterial community were primarily correlated with the pH, TN, NO_3_^−^-N, and NH_4_^+^-N, with Proteobacteria exhibiting the most pronounced correlation with these environmental factors, followed by Acidobacteria and Chloroflexi.

To further validate these findings, we conducted a Mantel test correlation analysis on the top 10 most abundant phyla (Figure 3B). The results confirmed that Proteobacteria had significant correlations with the pH, TN, and NO_3_^−^-N, particularly with the pH and TN. Other phyla also showed significant correlations with the environmental factors, with Acidobacteria having the strongest correlation, followed by Actinobacteria. These analysis results suggest that these bacterial phyla share similar nutritional preferences and may potentially engage in nutrient competition.

### 3.3. Bacterial Community Co-Occurrence Networks Are More Complex and Stable

To investigate the impact of reduced nitrogen fertilizer application on the network interactions within the soil microbial communities, a co-occurrence network analysis of the soil bacterial communities was performed using the OTUs. As depicted in Figure 4, the application of different levels of nitrogen fertilizer significantly influenced the characteristics of the soil bacterial co-occurrence networks. With an decrease in nitrogen fertilizer application, there was a noticeable increase in the number of nodes, total links, average degree, average weighted degree, and average clustering coefficient of the networks (Appendix A). This trend suggests that as the selective pressure from nitrogen fertilizer decreases, the network structure of the soil microbial communities tends to become more complex. Additionally, although reducing the application of nitrogen fertilizer decreased the modularity and number of modules (groups of highly interconnected nodes) of the network, it increased the stability of the network (Figure 4D, Appendix A).

### 3.4. Reduced Nitrogen Fertilizer Application Alters Soil Bacterial Community Network Composition

Following the reduction of nitrogen fertilizer application, the composition of the bacterial community networks underwent significant changes (Figure 5A). While certain taxa remained constant, many new species were integrated into each community network, replacing existing nodes and preserving the network structure (Figure 5B). This led to an increase in the dissimilarity index of the bacterial community networks after nitrogen reduction (Appendix A). The diminished supply of nitrogen resulted in a decline in the status of previously dominant bacterial taxa that supported the network, while some new taxa with stronger adaptive capabilities emerged as new pillars of the network (Figure 5B,C).

Furthermore, while positive links (co-occurrence relationships) dominated in all the bacterial co-occurrence networks, the proportion of negative links (competitive relationships) significantly decreased under low nitrogen conditions (Appendix A). This finding implies that reducing the nitrogen input may help alleviate competitive pressures among soil bacterial species, potentially influencing the structure and stability of bacterial communities. Further analysis of the top ten modules within the network revealed that under the influence of reduced nitrogen fertilizer, the modules within the network became larger and contained a more diverse range of bacterial species (Figure 4 and Appendix A). In addition, the associations between different modules also increased with the reduction of nitrogen (Appendix A). This indicates that in an environment with more limited resources, communities may develop more mutualistic relationships to collaboratively face the challenges posed by resource constraints.

## 4. Discussion

### 4.1. Competition Release among Bacteria Facilitates Diversification of Soil Bacterial Communities

Understanding the interactions between bacterial communities and nitrogen fertilizers is crucial for gaining a deeper insight into soil biogeochemical processes and the impact of the nutrient supply on soil crop productivity [39,40]. This study reveals that reducing the application rate of nitrogen fertilizer not only significantly increases the diversity of bacterial communities but also enhances their evenness and inter-community heterogeneity. This finding suggests that long-term nitrogen fertilizer application may be detrimental to maintaining soil bacterial diversity, which contrasts with some previous studies that indicated, despite changes in the soil microbial community structure due to nitrogen addition, the impact on microbial diversity was relatively minor [41].

The changes in the soil bacterial communities may be closely related to alterations in the soil physicochemical properties induced by the nitrogen fertilizer levels [42]. N is a limiting nutrient in most terrestrial ecosystems, and thus, the application of nitrogen fertilizer can promote microbial activities by alleviating nitrogen limitation [43,44]. However, microbes may exhibit preferences for certain nutrients, and an increase in the specific nutrient content may enhance the abundance of particular microbial communities [45]. This implies that while nitrogen fertilizer application can increase the overall microbial activity, this increase may not be uniform but rather biased toward microbial species that are particularly sensitive or preferential to nitrogen [46]. Proteobacteria are often considered eutrophic microbial groups and previous studies have shown that nitrogen fertilizer application typically promotes an increase in the abundance of this taxa [47,48]. This study further confirms the significant correlation between the abundance of Proteobacteria and the level of nitrogen fertilizer application. Conversely, when the application rate of nitrogen fertilizer is reduced, the nitrogen level in the soil decreases, which may lead to a reduction in the population of nitrogen-sensitive bacterial species. This reduction may weaken the monopolization of resources by specific bacterial populations, providing more living space and resources for other types of bacteria, thereby promoting an increase in the community diversity and evenness [49].

Our results support this view, showing that after reducing the application of nitrogen fertilizer, the abundance of the dominant Proteobacteria in the soil significantly decreased, while the abundance of other phyla, such as Acidobacteria, significantly increased. Competition for nutrients is prevalent among soil bacterial communities [50]. After reducing nitrogen fertilizer, the resource limitation in the soil intensifies, and nitrogen-dependent microbial populations (such as Proteobacteria) may decrease, creating ecological niches in the soil bacterial community [47]. The release of competition for nutrients brought by Proteobacteria provides a competitive advantage for other microbial populations with lower nitrogen requirements or those that can more effectively utilize other resources (such as Acidobacteria), leading to an increase in the abundance of these populations and occupation of the vacant ecological niches [48]. These results indicate that reducing the nitrogen fertilizer level significantly affects the composition and abundance of the soil bacterial communities, promoting the increase and decrease of specific microbial populations by altering the availability of nitrogen resources, which is also validated by the analysis of the structural equation models (Appendix A).

### 4.2. Competitive Release among Bacteria Promotes More Complex and Stable Bacterial Community Co-Occurrence Networks

The reduction in nitrogen fertilizer application has had a significant effect on the co-occurrence networks within the soil bacterial communities, an effect that can be explained by the mechanism of competitive release. With the decreased input of nitrogen, bacterial populations sensitive to nitrogen, such as those within the Proteobacteria phylum, have seen a substantial decrease in abundance. This decrease has led to the liberation of resources and ecological niches, allowing for the proliferation and thriving of other bacterial taxa within the community. Such changes have resulted in the increased complexity of the bacterial community network, with a greater number of nodes (species) and edges (interactions), as well as a rise in the average clustering coefficient.

In nitrogen-reduced environments, the bacterial community transitions from being dominated by nitrogen-dependent populations to a more diverse array of bacterial taxa. Emerging bacterial groups not only integrate into the existing network but also play a crucial role in maintaining and enhancing the network’s structure. New species more effectively utilize resources, filling the ecological niches vacated by nitrogen-sensitive species, thereby further promoting the complexity and stability of the network [51,52]. Under conditions of low nitrogen, the prominence of positive connections (co-occurrence relationships) within the co-occurrence network further emphasizes the importance of mutualistic symbiosis. The reduction in the nitrogen fertilizer input leads to a significant decrease in the negative connections (competitive relationships), indicative of diminished competitive pressure. The alleviation of competitive stress fosters additional cooperative interactions, thereby enhancing the overall stability of the bacterial community [53]. The presence of larger modules and increased bacterial species diversity, as well as the enhanced inter-connectivity between different modules, suggests that bacterial communities may respond to challenges posed by resource limitations by strengthening cooperative relationships [54,55]. Such cooperative relationships are essential for community resilience, as they buffer environmental stress and enhance community stability [56,57].

In summary, the phenomenon of competitive release induced by the reduction in the nitrogen fertilizer level contributes to the stability of soil bacterial co-occurrence networks by increasing the community diversity and complexity. The reduction in the nitrogen input not only eases competitive pressures but also enhances cooperative interactions among bacteria, thus promoting a more resilient and sustainable soil ecosystem [4,58].

## 5. Conclusions

This study demonstrates that reducing the nitrogen fertilizer application significantly enhances the diversity and stability of the soil bacterial communities, potentially leading to healthier soil ecosystems and more sustainable agricultural practices. Decreased nitrogen inputs reduce the dominance of nitrogen-sensitive bacterial taxa, thereby facilitating the emergence of a more diverse bacterial population. This shift leads to more complex and cooperative interactions within bacterial co-occurrence networks. These changes emphasize the role of competitive release in reshaping community structures and underscore the importance of sustainable nitrogen management strategies for maintaining ecological balance and improving soil health.

## Figures and Tables

**Figure 1 microorganisms-12-01434-f001:**
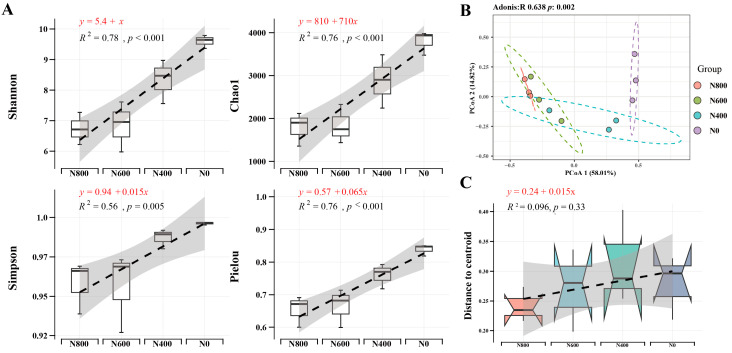
Changes in the soil bacterial community α- and β-diversity, and community heterogeneity following reduced nitrogen fertilizer application. (**A**) Changes in the α-diversity of bacterial communities, including the Shannon, Chao1, Simpson, and Pielou indices. (**B**) β-diversity of bacterial communities. (**C**) Heterogeneity of bacterial communities. N800: N fertilization rate of 800 kg/ha; N600: 600 kg/ha; N400: 400 kg/ha; N0: no N fertilization.

**Figure 2 microorganisms-12-01434-f002:**
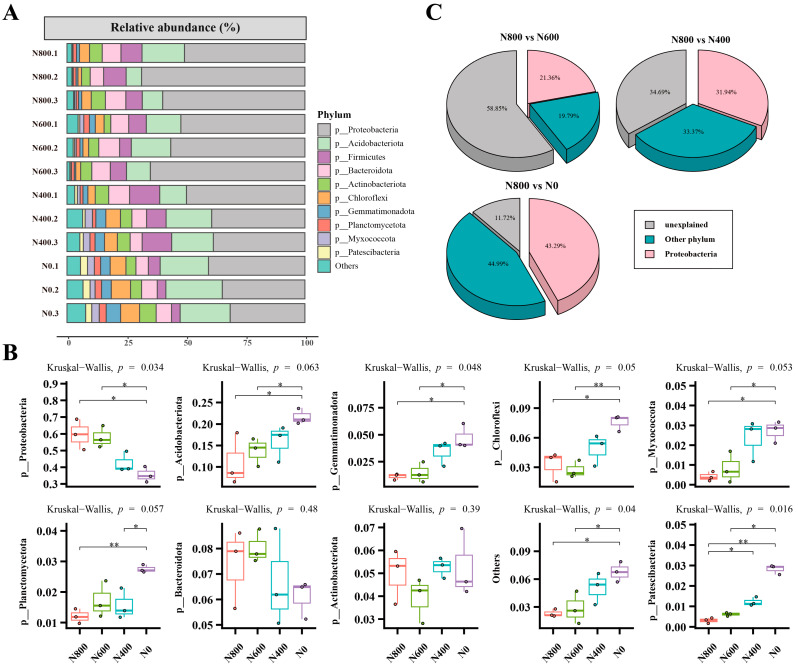
Changes in the soil bacterial community composition following reduced N fertilization. Stacked bar chart of abundance for the top 10 phyla (**A**) and their significance test (**B**). Contributions of different bacterial taxa to variations in the bacterial community composition (**C**). N800: N fertilization rate of 800 kg/ha; N600: 600 kg/ha; N400: 400 kg/ha; N0: no N fertilization. Significant differences (** *p* < 0.01, * *p* < 0.05).

**Figure 3 microorganisms-12-01434-f003:**
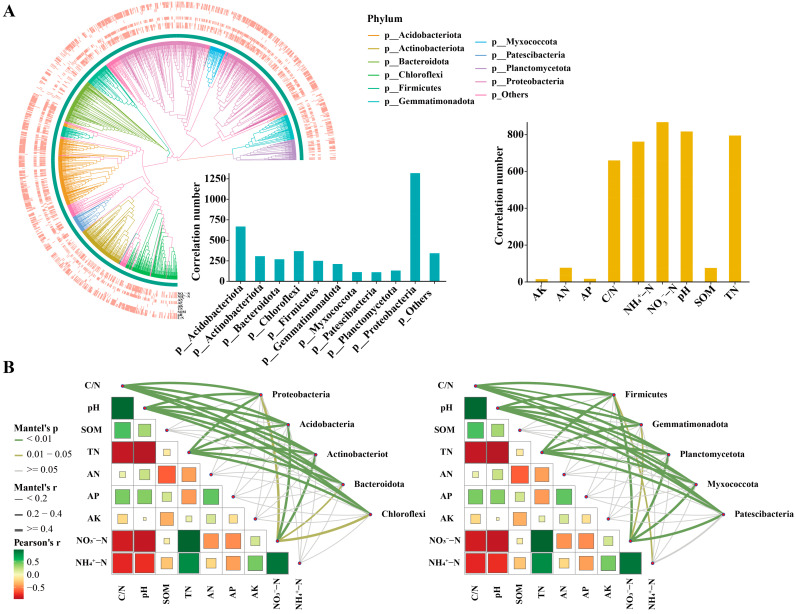
Correlation between the soil bacterial community’s main phylum and environmental factors following N fertilizer reduction. (**A**) Environmental preferences of bacterial communities; the circular diagram represents the phylogenetic tree of bacterial communities and their preferences for environmental factors. The two bar charts, respectively, show the number of correlations between major bacterial phyla and environmental factors, and the number of correlations between specific environmental factors and major bacterial phyla. (**B**) The Mantel test for major bacterial phyla and environmental factors; the thickness and color of the lines indicate the strength of the correlations.

**Figure 4 microorganisms-12-01434-f004:**
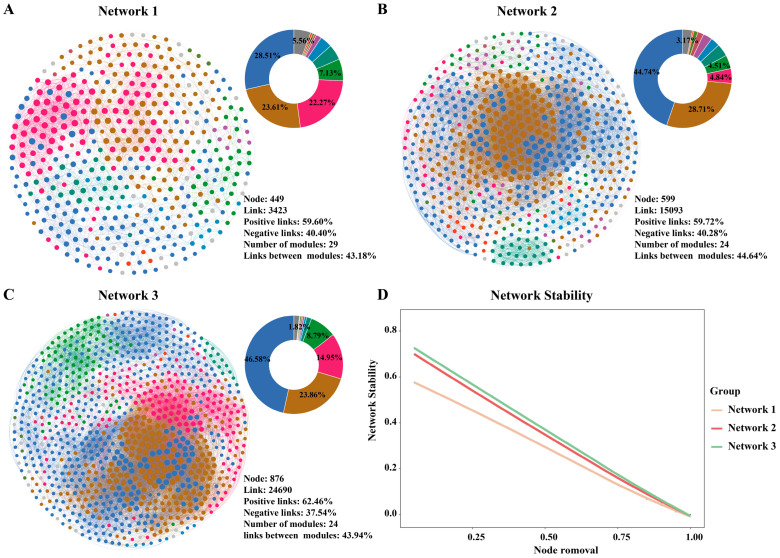
Changes in the co-occurrence networks and network stability of bacterial communities after nitrogen fertilizer reduction. Network 1, including N800 and N600 treatments (**A**); Network 2, including N600 and N400 treatments (**B**); Network 3, including N400 and N0 treatments (**C**); Network stability (**D**). Different colors represent different modules in the network; the pie chart represents the proportion of each module in the network.

**Figure 5 microorganisms-12-01434-f005:**
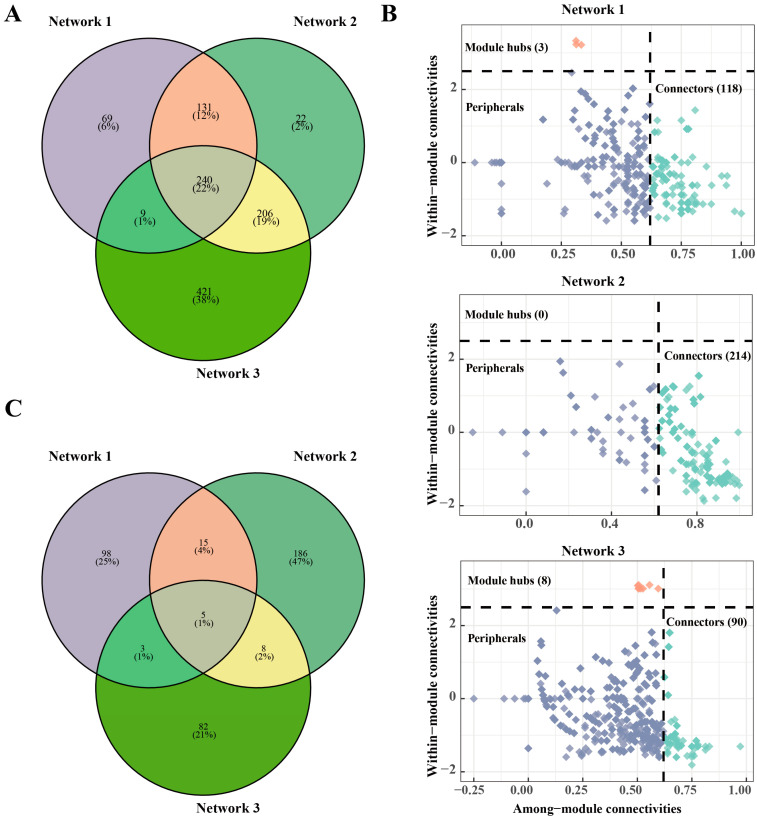
Changes in the network composition after nitrogen fertilizer reduction. Differences in the network composition based on the OTU level (**A**); Zi-Pi plots of the bacterial communities based on the OTU topological roles in the networks (**B**). Differences in the Zi-Pi composition across different networks (**C**). The threshold values of Zi and Pi for categorizing the OTUs were 2.5 and 0.62, respectively.

## Data Availability

The genomic sequencing data used in this study are available on NCBI Sequence Read Archive (SRA) under the accession number SRR29821674 to SRR29821685.

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
