# Peer review of "Reducing Application of Nitrogen Fertilizer Increases Soil Bacterial Diversity and Drives Co-Occurrence Networks"

_microorganisms, 2024, doi:10.3390/microorganisms12071434_

Round 1
Reviewer 1 Report
Comments and Suggestions for Authors
I recommend the figures use the N treatments in the axis in the lower to higher, left to right, usual orientation, and if possible, with the distances between points reflecting the actual values (i.e., with the distance between N0 and N400 double the distance between remaining points). All equations in Figure 1 are very problematic since the lines indicate decreasing values for the diversity indexes with increased N levels, but the equations indicate the opposite.
As for Figure 2A, I recommend changing it into a bar graph since the use of lines implies a continuity between the proportions of each class from N-level to N-level, which is clearly not the case since the figure has blanks between the N-levels.
Overall though, I find the paper quite interesting, with strong conclusions.
Comments on the Quality of English LanguageI recommend some minor language editing, for example, lines 110-121, which read as an instruction manual rather than a description of what was actually done in the experiment.
Another point to review is that some paragraphs are excessively long, such as the paragraph from line 272 to 312 (40 lines long). This markedly reduces reader comprehension. This paragraph, for instance, discusses several different topics, and would be easier to understand if broken along the topic lines.
Overall, though, the language is ok.
Author Response
Comment 1: I recommend the figures use the N treatments in the axis in the lower to higher, left to right, usual orientation, and if possible, with the distances between points reflecting the actual values (i.e., with the distance between N0 and N400 double the distance between remaining points). All equations in Figure 1 are very problematic since the lines indicate decreasing values for the diversity indexes with increased N levels, but the equations indicate the opposite.
Response 1: Thank you for your suggestions regarding our figure presentation. We understand your recommendation to arrange the nitrogen treatments in ascending order from left to right, with distances between points reflecting actual values. However, the aim of this study is to investigate the impact of varying levels of nitrogen reduction on soil bacterial communities. Therefore, we have opted to arrange the nitrogen treatments in descending order to more intuitively illustrate the effects of reducing nitrogen application. The current figure arrangement may more clearly show the trends and changes associated with reduced nitrogen application, helping readers better understand the study results. Regarding your suggestion on adjusting point distances to reflect actual values, we attempted this and found that it compromised the overall aesthetics and readability. Thus, after careful consideration, we decided to retain the original chart presentation. We believe the current presentation effectively conveys the research findings and adheres to scientific presentation standards. We hope you understand our choice. Concerning the mathematical formula mentioned in the chart, we would like to clarify: as nitrogen levels increase, diversity index values decrease, and conversely, as nitrogen levels decrease, diversity index values increase. The chart shows the trend of increasing diversity index values with decreasing nitrogen levels, thus aligning the mathematical formula with the figure presentation. We greatly appreciate your valuable feedback, which allowed us to review and confirm the accuracy and appropriateness of our research content. If you have further suggestions, we are more than willing to continue the discussion.
Comment 2: As for Figure 2A, I recommend changing it into a bar graph since the use of lines implies a continuity between the proportions of each class from N-level to N-level, which is clearly not the case since the figure has blanks between the N-levels.
Response 2: Thank you for your constructive and helpful suggestion. We have changed the species composition to a bar chart in the revised manuscript.
Comment 3: I recommend some minor language editing, for example, lines 110-121, which read as an instruction manual rather than a description of what was actually done in the experiment.
Response 3: Thank you for your constructive and helpful suggestion. We are sorry for the poor presentation in this section. We have rewritten this sentence according on your suggestions.
Comment 4: Another point to review is that some paragraphs are excessively long, such as the paragraph from line 272 to 312 (40 lines long). This markedly reduces reader comprehension. This paragraph, for instance, discusses several different topics, and would be easier to understand if broken along the topic lines.
Response 4: Thank you for your valuable feedback. Regarding the lengthy paragraph from lines 272 to 312 (40 lines in total) that you mentioned, we have split it into multiple paragraphs along thematic lines. Each paragraph now focuses on a single topic to improve the clarity and readability of the manuscript. Additionally, we have also divided other excessively long paragraphs throughout the manuscript. We hope these modifications will help readers better understand the content.
Reviewer 2 Report
Comments and Suggestions for Authors
The reviewed research has significant practical significance regarding the use of nitrogen fertilization. They emphasize the impact of nitrogen fertilization on bacterial communities and soil health. The section on methods requires some explanation.
Line 93: Please explain the principles on which the doses of nitrogen fertilizers were determined
Line 177, Figure 1: Please explain the abbreviations N400, N600, N800 in the caption under Figure 1; the same Line 192, Figure 2
Author Response
Comment 1: The reviewed research has significant practical significance regarding the use of nitrogen fertilization. They emphasize the impact of nitrogen fertilization on bacterial communities and soil health. The section on methods requires some explanation.
Response 1: Thank you very much for your approval of our research, particularly regarding the practical significance of nitrogen fertilization on bacterial communities and soil health. We highly value your comments and have made revisions based on your suggestions. Detailed explanations have been added to the methods section in the manuscript.
Comment 2: Line 93: Please explain the principles on which the doses of nitrogen fertilizers were determined.
Response 2: Thank you for your constructive and helpful suggestion. In the manuscript, we have included the following details: N800 represents the study site with the conventional nitrogen fertilization rate, N600 denotes a 25% reduction in nitrogen fertilization, N400 indicates a 50% reduction in nitrogen fertilization, and N0 corresponds to a 100% reduction in nitrogen fertilization.
Comment 3: Line 177, Figure 1: Please explain the abbreviations N400, N600, N800 in the caption under Figure 1; the same Line 192, Figure 2.
Response 3: Thank you for your valuable feedback. We have added explanations for the abbreviations in the captions under Figure 1 and Figure 2. We appreciate your attention to detail and believe these additions will enhance the clarity of our figures.
Reviewer 3 Report
Comments and Suggestions for Authors
· Consider changing the title of Manuscript in some more simple way. eg. Reducing the application of nitrogen fertilisers increases soil bacterial diversity and drives co-occurrence networks.
· Line 12 – The sentence is not clear enough, rephrase it.
· Line 22-23 – Merge this sentence with the sentence in line 14-15.
· Line 27 – consider adding keywords such as bacterial diversity and soil preservation
· Line 49 – is “proliferate” the right term?
· Line 84 – A four year field experiment (add the year when the experiment was set up and ended)
· Line 88 – is there any reference for determination of the soil type, are these some previous results of the authors?
· Line 93 – Did you apply only one dosage of fertilizers?
· Line m95 – what other treatments?
· Section 2.2 –Add space before brackets. Clarify were these analyses done for each sample for which the genetic analyses were done.
· Line 110 – The sentence is not clear enough, rephrase it.
· Section 2.3 – This section is not written in appropriate way, it seems as it has been copied from manufactures’ instruction. Rewrite it in a way you did for Section 2.4.
· Results section – where are the results for Section 2.2 presented? I suggest adding a table for these results; it can be shown as supplementary material.
· Figures 3 and 4 are too small, make it bigger (Figure 5 is good size).
· Line 369 – delete “please add”
Comments on the Quality of English Language
Minor editing of English language required.
Author Response
Comment 1: Consider changing the title of Manuscript in some more simple way. eg. Reducing the application of nitrogen fertilizer increases soil bacterial diversity and drives co-occurrence networks.
Response 1: Thank you for your constructive and helpful suggestion. According to your suggestion, in the revised manuscript, for the title was changed to "Reducing the application of nitrogen fertilizer increases soil bacterial diversity and drives co-occurrence networks".
Comment 2: Line 12 – The sentence is not clear enough, rephrase it.
Response 2: Thank you for your constructive and helpful suggestion. We are sorry for the poor presentation in this section. We have rewritten this sentence.
Comment 3: Line 22-23 – Merge this sentence with the sentence in line 14-15.
Response 3: Thank you very much for your reminder. The revised manuscript has made modifications to these sentences.
Comment 4: Line 27 – consider adding keywords such as bacterial diversity and soil preservation.
Response 4: Thank you for your constructive and helpful suggestion. We have added the keywords in the revised manuscript.
Comment 5: Line 49 – is “proliferate” the right term?
Response 5: Thank you very much for your feedback and suggestions. After careful consideration, we have decided to retain the term "proliferate." We conducted an in-depth review of the relevant literature and noted that the term "bacterial proliferation" has been widely adopted in numerous articles, and it aligns with the intended meaning in our manuscript. Therefore, we will continue to use this term in our paper. We greatly appreciate your valuable advice, which gave us the opportunity to review and confirm our terminology to ensure its accuracy and appropriateness.
Comment 6: Line 84 – A four year field experiment (add the year when the experiment was set up and ended)
Response 6: Thank you for your reminder. We have added the start and end years of the experiment in the revised manuscript.
Comment 7: Line 88 – is there any reference for determination of the soil type, are these some previous results of the authors?
Response 7: Thank you very much for your reminder. Soil types in this study were determined with reference to relevant literature and have been added in the revised manuscript.
Comment 8: Line 93 – Did you apply only one dosage of fertilizers? Line 95 – what other treatments?
Response 8: Thank you very much for your valuable comments. We apologize for not describing the details clearly in the manuscript. In this study, we used urea as the nitrogen source to adjust the nitrogen fertilizer application rates in different treatments. Specifically, urea was applied prior to each planting. While the amount of nitrogen applied varied across treatments, all other agronomic practices were consistent to ensure comparability between treatments. Additionally, to ensure nutritional balance, the same doses of phosphorus and potassium fertilizers were applied in each treatment. This was specifically considered in the experimental design to prevent plant growth from being limited by a single nutrient element. We have added this detailed information to the revised manuscript to provide a more complete experimental background and methodology for our readers. The relevant content has been included in both the main text and supplementary materials, with the hope that it clearly explains the specific operations of the experiment and its scientific basis.
Comment 9: Section 2.2 –Add space before brackets. Clarify were these analyses done for each sample for which the genetic analyses were done.
Response 9: Thank you for pointing out this mistake. In the revised manuscript, we have added spaces before the brackets. We have also rewritten the sentence according on your suggestions.
Comment 10: Line 110 – The sentence is not clear enough, rephrase it.
Response 10: Thank you for your constructive and helpful suggestion. We have rewritten this sentence.
Comment 11: Section 2.3 – This section is not written in appropriate way, it seems as it has been copied from manufactures’ instruction. Rewrite it in a way you did for Section 2.4.
Response 11: Thank you for your constructive and helpful suggestion. We are sorry for the poor presentation in this section. We have rewritten this sentence according on your suggestions.
Comment 12: Results section – where are the results for Section 2.2 presented? I suggest adding a table for these results; it can be shown as supplementary material.
Response 12: Thank you for your reminder. These results are presented in table form in the supplementary materials.
Comment 13: Figures 3 and 4 are too small, make it bigger (Figure 5 is good size).
Response 13: Thank you for your suggestion, we have made revisions to figures 3 and 4.
Comment 14: Line 369 – delete “please add”.
Response 14: Thank you for pointing out this mistake, revised in the returned manuscript.
Comment 15: Minor editing of English language required.
Response 15: Thank you for your valuable feedback. We appreciate your comments on the need for minor English language editing. In revising the manuscript, we carefully reviewed the manuscript again and made the necessary adjustments to ensure clarity and correctness of the language. We aim to improve the readability and professional quality of our text. Thank you once again for your helpful suggestions.
Round 2
Reviewer 3 Report
Comments and Suggestions for Authors
Corrections are done in a good way.